# Bathymetry Determination Based on Abundant Wavenumbers Estimated from the Local Phase Gradient of X-Band Radar Images

**Laurence Zsu-Hsin Chuang [1], Li-Chung Wu [2],\* , Yung-Da Sun [3] and Jian-Wu Lai [4]**

[1] Institute of Ocean Technology and Marine Affairs, National Cheng Kung University, Tainan 701, Taiwan; zsuhsin@ncku.edu.tw
[2] Coastal Ocean Monitoring Center, National Cheng Kung University, Tainan 701, Taiwan
[3] Naval Meteorological and Oceanographic Office, Kaohsiung 813, Taiwan; g9114072005@mail.cnmoo.mnd.gov.tw
[4] Marine Industry and Engineering Research Center, National Academy of Marine Research, Kaohsiung 813, Taiwan; laijw@namr.gov.tw
\* Correspondence: jack18@mail.ncku.edu.tw

**Abstract:** A phase gradient (PG)-based algorithm is proposed in this study to determine coastal bathymetry from X-band radar images. Although local wavenumbers with the same spatial resolution of the wave field can be obtained from the wave field using the PG method, only a single wavenumber result can be extracted from each location theoretically. Due to the influence of unavoidable noise on the wave field image, single wavenumber estimation often shows high uncertainty. This study combines a bandpass filter and directional pass filter to produce different nearly monocomponent wave fields from X-band radar images and then estimates more wavenumbers from these wave fields using the PG method. However, the distributions of wavenumbers in higher-frequency bins still show high variance because the strength of wave signals is weak. We confirmed that the uncertain wavenumber–frequency pairs can be improved using the Kalman filter and are more consistent with the dispersion relation curve. To decrease the influence of inaccurate wavenumbers, we also use the strength of the wave signals as the weights for the least-squares fit. Although the depth errors from shallow-water areas are still unavoidable, we can remove the inaccurate depth estimation from shallow-water areas according to the coefficients of determination of the fitting. In summary, the algorithm proposed in this study can obtain a bathymetry map with high spatial resolution. In contrast to the depth result estimated using a single wavenumber of each frequency bin, we confirm that more wavenumbers from each of the frequency bins are helpful in fitting the dispersion relation curve and obtaining a more reliable depth result.

**Keywords:** wave dispersion relation; directional pass filter; nearly monocomponent wave fields

## 1. Introduction

Coastal bathymetry is key to supporting different maritime and coastal applications, such as the safety of ship navigation and assessment of coastal risks. Underwater acoustic techniques are undoubtedly the most popular way to obtain high-precision depth data [1–3]. However, the risk due to sea states and the time consumption often increase the difficulty of in situ operation. Remote sensing provides potential ways to obtain coastal bathymetric information more efficiently. Both optical video and radar images have been proven to be feasible for determining coastal bathymetry [4–10]. Video image sequences, whose temporal and spatial resolutions are better than those of radar image sequences, are helpful in determining the water depth in shallower sea areas. However, the range of bathymetry determination using X-band radar images can be much larger, up to several kilometers. Nautical X-band radar is normally used to detect coastlines and obstacles on the sea surface on board a ship. Studies have proven its feasibility for presenting the patterns of ocean

waves since the late 1970s [11,12], and X-band radar is currently one of the most popular tools for ocean remote sensing.

The wave dispersion relation of the linear theory for surface gravity waves, which describes the wavenumber–frequency features of gravity waves at different water depth conditions, is the most common way to estimate bathymetry from remotely sensed images. Because the wavenumber–frequency structure can be obtained from the spatial–temporal wave patterns in sea surface image sequences [4], different algorithms have been proposed to extract wavenumber–frequency information accurately and efficiently.

Regardless of whether wave pattern image analysis is implemented in the spatial-temporal domains or in the spectral domains [5,13,14], the window size of the image is always the result of a tradeoff between the accuracy of the estimates and the homogeneity constraints on the sea parameters [15]. Because the fast Fourier transform algorithm reduces the number of computations, an increasing number of studies have focused on depth determination from the spectral domain. The wavenumber obtained from the image spectrum is a popular and reliable method. However, the precise wavenumber result comes from the high-resolution image spectrum, which necessitates a spectral transform from the large window size of the image. When large wave pattern images are analyzed, the inhomogeneity within coastal images is often unfavorable for obtaining bathymetry maps with high spatial resolution.

In addition to image analysis with a specific window size, another category of bathymetry retrieval methods estimates the wavenumber from the local phase gradient (PG) within a monofrequency image and then calculates the wavenumber–frequency pair [16,17]. The PG over the Fourier transform technique allows higher spatial resolution and computational efficiency [18]. A depth result can be determined from one wavenumber–frequency pair using the dispersion relation. However, errors of local phase estimation under the joint influences of the imaging mechanism, observation noise, and multiple wave frequencies are unavoidable [19]. As a result, error in depth determination using a single wavenumber–frequency pair that is estimated from sea surface images is inevitable. Although the water depth value can be determined by fitting the dispersion relationship curve using all frequency-dependent wavenumbers, the derived wavenumbers for each frequency bin using the PG method are still uncertain. If a sequence of records is processed, we can average the bathymetry maps to reduce noise or implement a Kalman filter to objectively update the bathymetry estimates [18,20]. However, both the average and Kalman filter methods rely on sufficient records with clear wave pattern images. In the case of limited image records, we need to implement another way to improve the accuracy of depth determination. Under the PG method, this study aims to estimate the plural local wavenumbers at each of the frequency bins to determine the bathymetry using marine radar images. To obtain abundant local wavenumbers at each location, spectral bandpass filtering and directional filtering are carried out on individual pixels in isolation. In subsequent sections, we will demonstrate the entire image processing procedure and confirm its practicability using coastal radar images.

## 2. Methods and Data

### 2.1. Theoretical Preliminaries

Similar to most studies on bathymetry determination using remotely sensed images, our study applies the theory of the wave dispersion relationship for depth estimation. The dispersion relation under different depth conditions is

$$\omega = \sqrt{gk\tanh(kd)} + \vec{\mathbf{k}} \cdot \vec{\mathbf{U}},\tag{1}$$

where $\omega$ is the angular frequency, $d$ is the water depth, and $g$ is acceleration due to gravity. $\vec{\mathbf{k}} = (k_x, k_y)$ and $\vec{\mathbf{U}}$ are the wavenumber and current vectors, respectively. $k$ is the wavenumber modulus. To simplify the depth estimation, we ignore the effects of wave–current interactions. Note that Equation (1) is based on the linear wave theory. Although

different nonlinear wave theories have been proposed, the practicality of employing the nonlinear wave dispersion relation in depth determination is still under discussion. In our study, we focus only on depth estimation using the linear wave dispersion relation.

Natural ocean waves can be seen as the sum of some statistically independent, harmonic waves propagating in different directions across the sea surface. From the point of view of the spectral domain, the directional spectrum represents the distribution of wave energy not only in the frequency domain but also in a direction [21]. As a result, remotely sensed images of the sea surface always present multiple components of wave patterns. To obtain reliable wavenumbers from the wave image signals using the PG method, we first have to estimate the local phases from nearly monocomponent images. Later, we will explain why we use nearly monocomponent images instead of pure monocomponent wave field images.

### 2.2. Methods

Figure 1 shows the procedure for obtaining nearly monocomponent images of the wave pattern. In our study, the original wave image signals were decomposed into different nearly monocomponent signals by bandpass and directional filtering. The image sequences $I(\vec{\mathbf{x}}, t)$ that present the image intensity at different spatial locations $\vec{\mathbf{x}} = (x, y)$ and times ($t$) are transformed into the spectral domains $S(\vec{\mathbf{k}}, \omega)$ using the three-dimensional, fast Fourier transform (FFT). Note that computing the temporal and spatial gradients of the phase can yield both the wavenumber and the frequency of the moving periodic structure directly [22]. However, both the wavenumber and the frequency are sensitive to noise from image sequences. The short-term characteristics of wave records can be reasonably treated as nearly stationary [23]. Subsequently, we will point out that the duration of our radar image sequences is 183 s, and ocean waves can be deemed stationary within this duration. As a result, the frequency information is extracted from the spectrum instead of the estimation of the temporal gradient of the phase.

Due to different radar imaging mechanisms [24], not only ocean wave signals but also nonwave components are included within the wave pattern image. To obtain the spectra with only ocean wave components $S_F(\vec{\mathbf{k}}, \omega)$, the wave dispersion shell is implemented here to filter out nonwave signals from $S(\vec{\mathbf{k}}, \omega)$ [25]. $S_F(\vec{\mathbf{k}}, \omega)$ still includes wave components with multiple frequencies, which are unfavorable for determining the local wavenumber using the PG method. We isolate different wave components using both a bandpass filter $F_B(\omega)$ and a directional filter $F_D(\theta)$ [16]. To obtain the signals with a monofrequency component, we preserve the power density at only one specific angular frequency bin $\omega_i$ for the bandpass filter $F_B(\omega)$:

$$F_B(\omega) = \begin{cases} 1 & \omega = \omega_i \\ 0 & \omega \neq \omega_i \end{cases}, \tag{2}$$

where the angular frequency bins $\omega_i$ are determined from the temporal length and interval of image sequences on the basis of the FFT algorithm. Although the period of gravity ocean waves can be up to 20 s, depth determination using wave components that are too long results in the issue of nonlinear waves, which is out of the scope of our study. On the other hand, the wave dispersion relation shows that depth determination using wave components that are too short is quite sensitive to the influence of ocean currents. To avoid wavenumber estimations from inappropriate frequency bins, we limited the range of $\omega_i$ for practical filtering. In our study, we limited $\omega_i$ to within the range of $2\pi/12$ (rad/s) to $2\pi/5$ (rad/s).

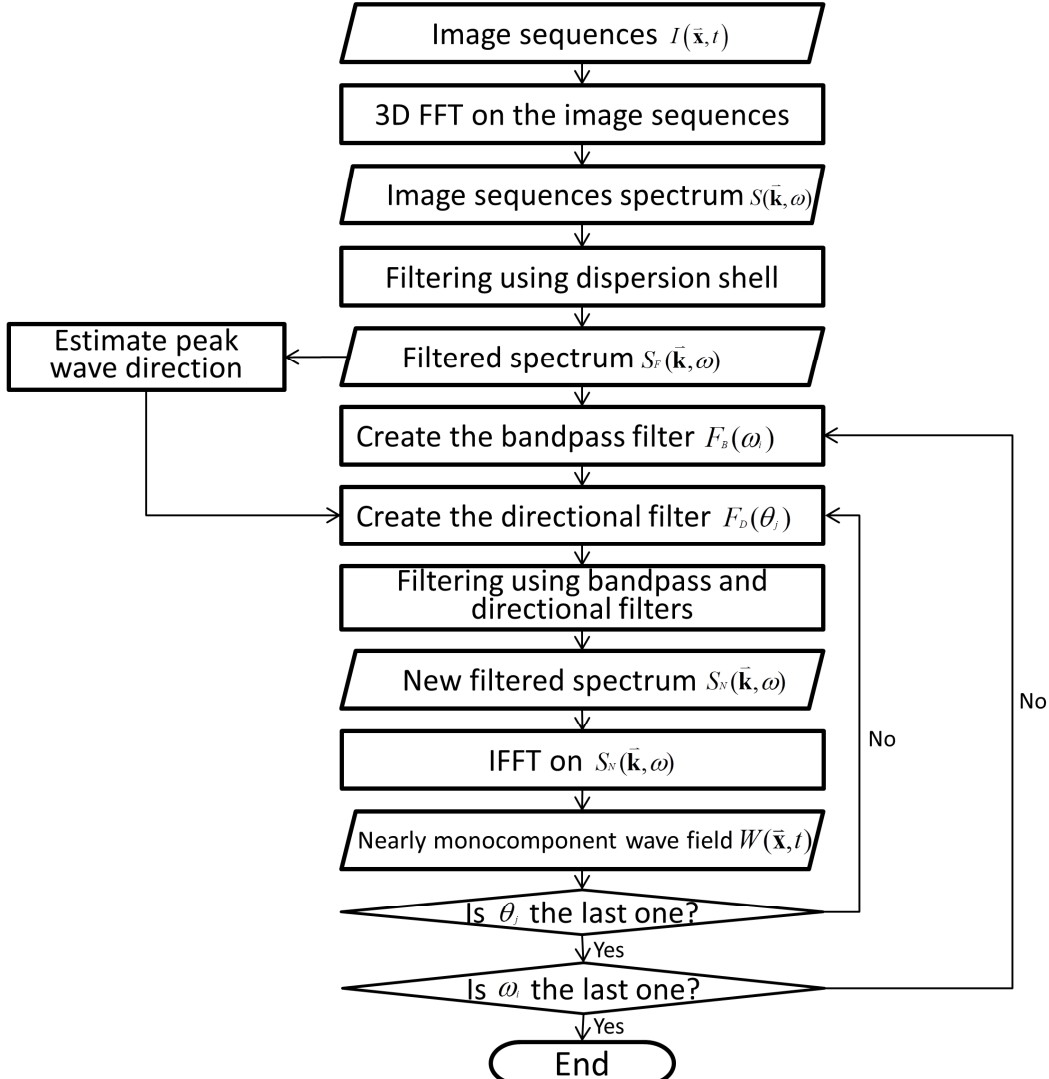

**Figure 1.** Flowchart for the image process in our study. Three different types of filters are applied here to obtain a nearly monocomponent wave field.

For directional filtering $F_D(\theta)$, we designed the filter based on the dominant wave direction $\theta_p$, which is defined as the direction of the most energetic wave in the spectrum according to the World Meteorological Organization:

$$F_D(\theta) = \begin{cases} 1 & \theta = \theta_p + \theta_j \pm (\Delta\theta/2) \\ 0 & otherwise \end{cases}, \tag{3}$$

$$\theta_j = -N, -N+1, \ldots, -1, 0, 1, \ldots, N-1, N, \tag{4}$$

In which $\theta_j$ is the key to obtaining more than one local wavenumber at a single pixel of the wave image. $\Delta\theta$ is the width of the directional pass filter. Although a small value of $\Delta\theta$ can produce a pure monocomponent wave field image, the pure monocomponent image presents only homogeneous wave features, which is unfavorable for determining inhomogeneous wavenumbers from the spatial domain. In the case of wave refraction in shallow water, a directional filter with a directional width that is too narrow also interferes with wavenumber estimation from refracted waves. To determine the wavenumber from a coastal image, we establish a larger value of $\Delta\theta$. In our study, $N = 15$ and $\Delta\theta = 30°$.

To apply directional filtering, $S_F(\vec{\mathbf{k}}, \omega)$ is transformed into the polar coordinates $S_F(k, \theta, \omega)$ [26]:

$$S_F(k, \theta, \omega) = k S_F(\vec{\mathbf{k}}, \omega). \tag{5}$$

After implementing bandpass and directional filtering, we can obtain a nearly mono-component wave field $W(\vec{\mathbf{x}}, t)$ from the filtered spectrum $S_N(\vec{\mathbf{k}}, \omega)$:

$$S_N(k, \theta, \omega) = S_F(k, \theta, \omega) F_B(\omega) F_D(\theta), \tag{6}$$

$$S_N(\vec{\mathbf{k}}, \omega) = (1/k) S_N(k, \theta, \omega), \tag{7}$$

$$W(\vec{\mathbf{x}}, t) = \text{IFFT}[S_N(\vec{\mathbf{k}}, \omega)], \tag{8}$$

where $W(\vec{\mathbf{x}}, t)$ is the result of complex numbers. The local phase $\phi(\vec{\mathbf{x}})$ can be extracted from $W(\vec{\mathbf{x}}, t)$ at different locations $\vec{\mathbf{x}}$:

$$\phi(\vec{\mathbf{x}}) = \tan^{-1}(\Im(W(\vec{\mathbf{x}})) / \Re(W(\vec{\mathbf{x}}))), \tag{9}$$

where $\Im(W(\vec{\mathbf{x}}))$ is the imaginary part of $W(\vec{\mathbf{x}})$ and $\Re(W(\vec{\mathbf{x}}))$ is the real part of $W(\vec{\mathbf{x}})$. For a radar image sequence with $T$ images, we can obtain $T$ phase results from the same location of $W(\vec{\mathbf{x}}, t)$ in theory. However, the estimated phase $\phi(\vec{\mathbf{x}}_n, t)$ is always a fixed value at the fixed location $\vec{\mathbf{x}}_n$ and different $t$ due to the implementation of the bandpass filter.

We estimate the components of the local wavenumber $k_x(\vec{\mathbf{x}}_n)$ and $k_y(\vec{\mathbf{x}}_n)$ by the gradient of the local phase along the horizontal direction and vertical direction, respectively. After implementing different directional filters, we obtain more wavenumbers at each of the given frequency bins. Note that the phase estimation is still sensitive to the noise within $W(\vec{\mathbf{x}}, t)$. Although different filters are applied to filter out nonwave components from remotely sensed images, we still observe some weak signals from the land areas within $W(\vec{\mathbf{x}})$, where the phase estimation is meaningless for the depth determination. To avoid phase estimations from locations with very weak wave signals or land areas, we remove the results of $W(\vec{\mathbf{x}})$ when the normalized magnitude $\widetilde{W}(\vec{\mathbf{x}}) = \left| W(\vec{\mathbf{x}}) \right| / W_M$ is less than 0.2, where $W_M$ is the maximal value of $\left| W(\vec{\mathbf{x}}) \right|$. For each frequency bin, we can obtain $2N + 1$ nearly monocomponent wave fields and then estimate $2N + 1$ local wavenumbers after repeatedly implementing the directional filter $F_D(\theta)$ using different $\theta_j$. However, the distribution of wavenumbers in each frequency bin can be disordered due to the influence of unavoidable noise. To improve the quality of the estimated wavenumbers, Kalman filtering is applied here. The Kalman filter is an estimator that is statistically optimal with respect to any quadratic function of estimation error [27]. Kalman filtering was carried out successfully in the time domain for depth determination over longer periods of time [18]. In our study, we focus on depth determination from single radar measurements using 128 image sequences.

After implementing each directional filtering step, we can obtain wavenumber–frequency pairs $(k_m, \omega_m)$ from different, nearly monocomponent wave fields. Because the frequencies that are obtained from the results of FFT are regular, we focus on the wavenumbers of Kalman filtering. On the basis of the Kalman filter algorithm, wavenumber $\hat{k}_m^-$ can be predicted using wavenumber $\hat{k}_{m-1}^-$:

$$\hat{k}_m^- = A_{m-1} \hat{k}_{m-1}^- + u_{m-1}, \tag{10}$$

$$P_m^- = P_{m-1} + Q, \tag{11}$$

in which $A_{m-1}$ is the transition matrix of the dynamic model. $u_{m-1}$ is the process noise at frequency step $m-1$. The subscripts $m$ and $m-1$ represent an adjacent pair of frequency bins. $P_m^-$ is the a priori estimate error covariance of the estimated $\hat{k}_m^-$. $Q$ is the process noise of the discrete model. Equations (10) and (11) describe the prediction steps of Kalman filtering. The update steps or corrected steps of Kalman filtering are:

$$\hat{k}_m = \hat{k}_m^- + G_m(k_m - \hat{k}_m^-), \tag{12}$$

$$P_m = P_m^-(1 - G_m), \tag{13}$$

$$G_m = \frac{P_m^-}{P_m^- + E}. \tag{14}$$

The Kalman gain $G_m$ shows the extent to which the predictions should be corrected in step $m$. $E$ is the measurement error covariance. The detailed procedure of the Kalman filter algorithm and the solution for $A_{m-1}$ and $Q$ are given by Hartikainen et al. (2011) [28].

The final depth estimate is the value that yields the best weighted least-squares fit of the dispersion relation curve using wavenumber estimations. The weights in our study are $\widetilde{W}(\overrightarrow{\mathbf{x}})$, which represents the normalized magnitude of wave signals at different spatial locations.

### 2.3. Study Area and the Data and Their Sources

Figure 2 shows the layout of our study site, which is located in the coastal area of Tainan, Taiwan. Within the monitoring area, we collected survey bathymetry data that were measured by a single-beam echo sounder (Teledyne Odom Hydrographic Hydrotrac 200 kHz) from a boat. The resolution of the survey measurement is 0.01 m. The accuracy of survey data is 0.01 m + 0.1% of depth value. The maximum depth range can reach 100 m, which is suitable for the depth measurement in our study area. The colors in Figure 2a represent the survey bathymetry. The deepest survey water depth was 14 m, and the survey area is a part of the whole radar monitoring area. The influences of tides on depth determination in coastal areas were considered in our study. The mean tidal range in the study site is 1.24 m. To compare the estimated depth from radar with the survey depth from the echo sounder, both the radar-estimated depths and the survey depth were corrected using simultaneous in situ tide data. Figure 2b presents the bathymetry profile at a fixed latitude. The nearshore slope is approximately 0.01.

Figure 2c shows the radar-observed images during our field experiment and presents the wave patterns in the northwest part of the image. There was a data buoy moored offshore approximately 8.5 km from the radar antenna. The significant wave height, peak wave period, dominant wave direction, near sea surface current, and direction measured from the data buoy were 1.69 m, 7.8 s, 348°, 0.34 m/s, and 40°, respectively, during our field experiment. The radar had a peak power output of 25 kW, operated at 9.4 GHz (X-band) with HH polarization, and was equipped with an antenna 2 m in length that had a horizontal beam width of 1.2°. The rotation rate of the radar antenna was 42 rpm, which yielded an image sequence sampling rate of 0.7 Hz. The radar measurement collected 128 continuous images, which took 183 s. We collected radar images over a spatial range of 3750 m with a grid size of 7.5 m. The system stored the logarithmically amplified radar backscatter information at a 12-bit image depth. Note that the radar and in situ measurements were not performed simultaneously. The survey depth measurements were performed 8 days after the radar observations.

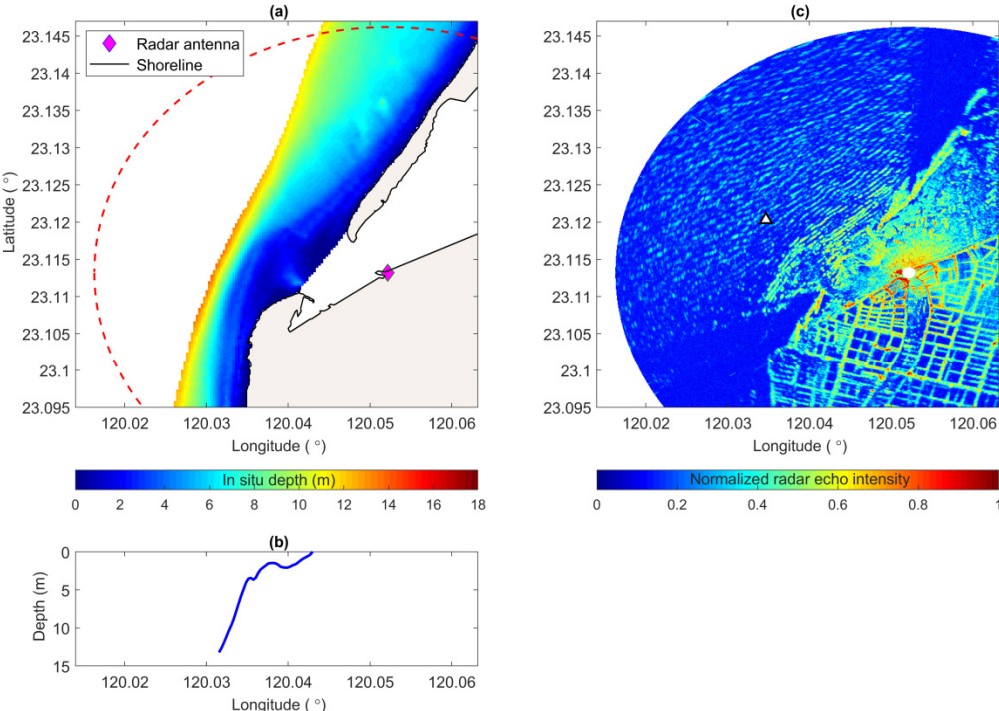

**Figure 2.** (**a**). Layout of our study site. The area of radar measurement is delineated by a red dashed line. The land area is marked in gray. The radar antenna is marked using a diamond near the mouth of the lagoon. (**b**). Bathymetry profile at a latitude of 23.115°. (**c**). Radar image of our experiment. The checkerboard-like patterns in the southeast area on the radar image are induced by radar echoes from fish farms. The area for depth determination is mostly in the northwest part, which presents clear patterns of ocean waves. The estimated local wavenumbers at the location marked by the triangle are presented in Figure 4.

## 3. Results and Discussion

Figure 3 shows the nearly monocomponent images of ocean waves after bandpass and directional filtering. For the image process of our study, there are 23 angular frequency bins ($\omega_i$) within the range of $2\pi/12$ (rad/s) to $2\pi/5$ (rad/s). Figure 3 shows the results of bandpass filtering using $\omega_i = 2\pi/7.9$, which is near the in situ peak wave angular frequency. The dominant wave direction $\theta_p$, which is estimated from radar images, is 313°. Figure 3a,b shows the results using the directional bands of $\theta = \theta_p + 15 \pm (\Delta\theta/2)$ and $\theta = \theta_p - 15 \pm (\Delta\theta/2)$, respectively. The wave patterns show different directions between these two figures.

Because we remove the results of $W(\vec{\mathbf{x}})$ when the normalized magnitude $\widetilde{W}(\vec{\mathbf{x}})$ is less than 0.2, some parts of the wave field are excluded due to weak wave signals. Even though we implemented 31 directional filters to produce 31 nearly monocomponent wave fields, we could not obtain all 31 wavenumbers at a specified angular frequency at locations where the wave signal was weak. Because sufficient wavenumbers are necessary to fit the dispersion relation curve with local depth, we skip the fitting process when the total number of wavenumber–frequency pairs is less than 300.

Figure 4 shows the abundant wavenumber–frequency pairs estimated at the fixed location of a triangle marker in the radar image of Figure 2c. Figure 4a shows the raw wavenumbers estimated directly from different nearly monocomponent wave fields. The dispersion relation curve using the survey depth of this specified location is also presented in the figure. Although parts of the raw wavenumber–frequency pairs are consistent with the dispersion relation curve, the raw wavenumbers show high variance in higher-frequency bins. This poor reliability from raw wavenumber–frequency pairs is obvious and therefore needs to be improved. Figure 4b shows the wavenumber–frequency pairs that are corrected using Kalman filtering. The Kalman-filtered wavenumber–frequency

pairs show much smaller variance and are more consistent with the dispersion relation curve. Figure 4 also shows that the deviated wavenumbers from high-frequency bins are mostly estimated from weak $\widetilde{W}(\vec{\mathbf{x}})$. Note that our study applies the magnitudes of wave signals $W(\vec{\mathbf{x}})$ as the weights for the least-squares fit of the dispersion relation curve.

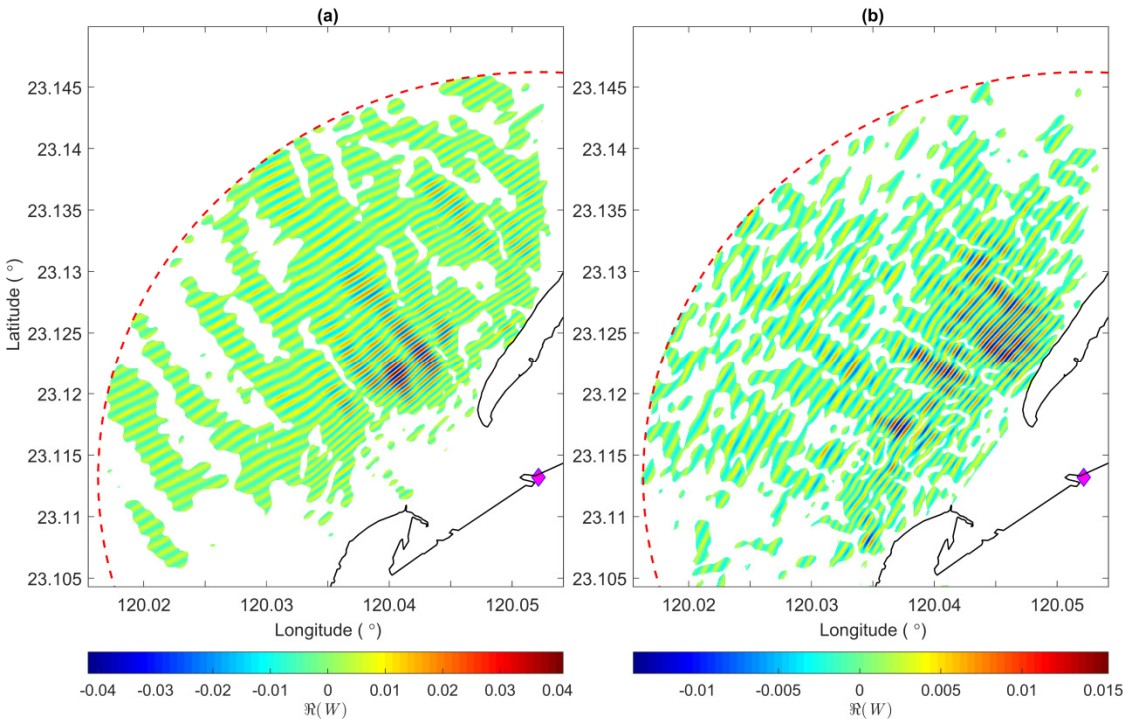

**Figure 3.** Results of $\Re(W(\vec{\mathbf{x}}))$ using the filters with the directional band: (**a**). $\theta = \theta_p + 15 \pm (\Delta\theta/2)$; (**b**). $\theta = \theta_p - 15 \pm (\Delta\theta/2)$. Parts of the area within the spatial domain of $\Re(W(\vec{\mathbf{x}}))$ are excluded when the strength is low ($\left|\widetilde{W}(\vec{\mathbf{x}})\right| < 0.2W_M$).

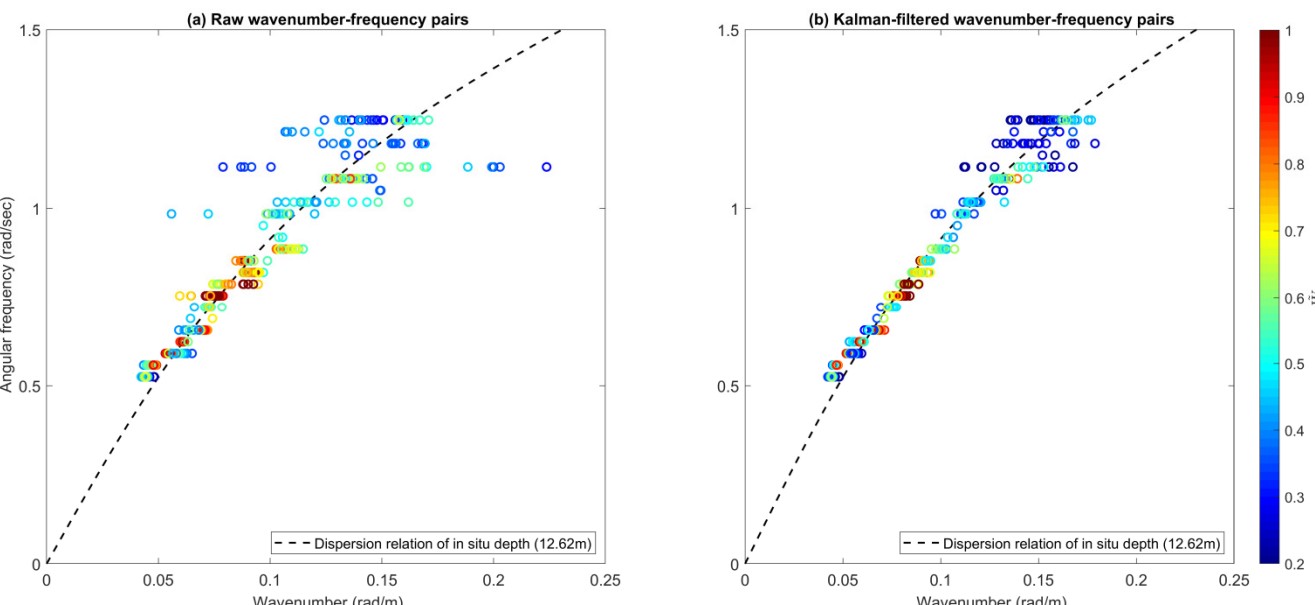

**Figure 4.** The relationship between estimated wavenumber–frequency pairs and the dispersion relation curve. (**a**). Raw wavenumber–frequency pairs; (**b**). Kalman-filtered wavenumber–frequency pairs. The colors of the wavenumber–frequency pairs indicate $\widetilde{W}$, which is represented by the strength of the nearly monocomponent wave field.

Figure 4 shows only one case at a specified location. We further discuss the features of the wavenumbers extracted from all locations within the wave field. As shown in Figure 5, the different upper whiskers show unclear connections with the frequencies. The normalized frequency $\widetilde{\omega}$ in Figure 5 is defined as:

$$\widetilde{\omega} = \omega_i / \omega_p, \tag{15}$$

where $\omega_p$ is the peak angular frequency of the ocean waves, which can be obtained from the filtered image spectrum. Figure 5 shows that all of the upper whiskers are smaller than 0.08 (rad/m). However, we observe that the distributions of the upper outliers are related to the wave frequency. From the cases of higher frequency bins, more inaccurate wavenumber estimations are revealed. This should be a reason that higher wave frequency bins are sensitive to the sea surface current $\overrightarrow{\mathbf{U}}$. We will discuss this issue in a subsequent section.

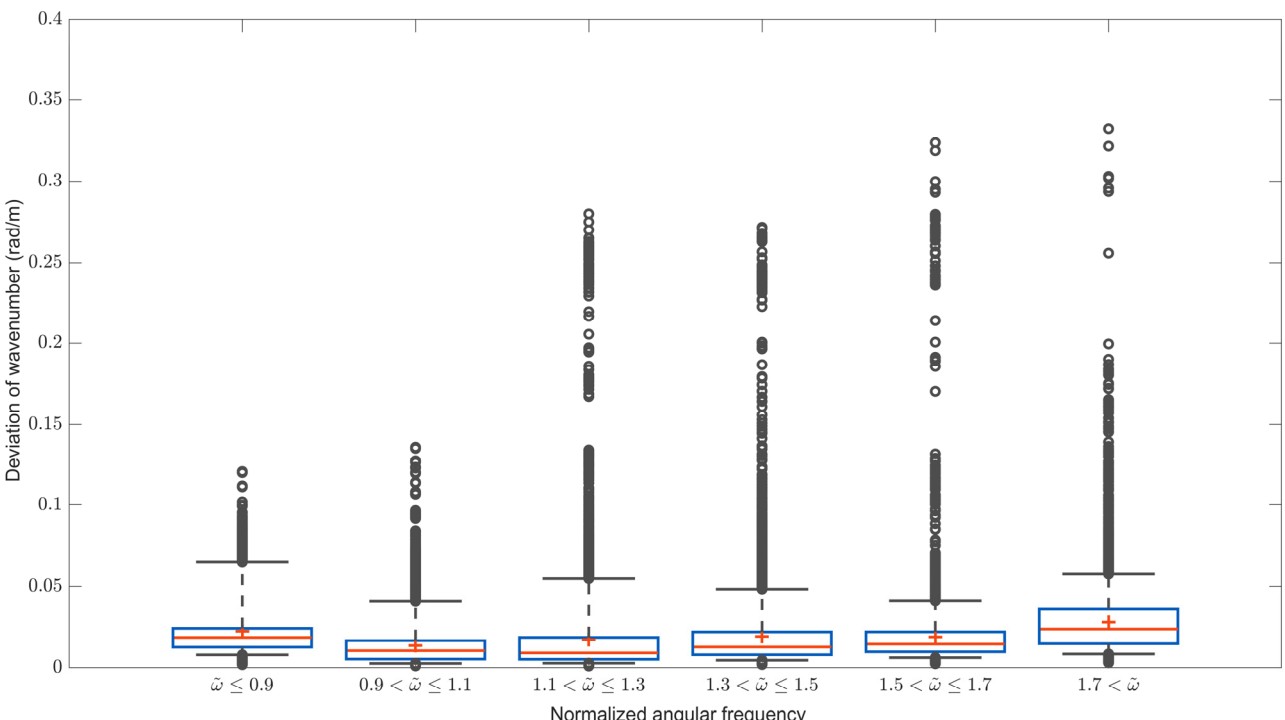

**Figure 5.** Deviations in the wavenumber at different normalized frequency conditions. The top and bottom of each box are the 75th and 25th percentiles of the samples, respectively. The line in the middle of each box is the sample median. The cross mark near the middle line is the sample mean. The upper and lower whiskers indicate the 100th and 0th percentiles of the samples, excluding any outliers, which are marked by circles.

As we mentioned above, the inaccurate wavenumbers are also due to the local phase gradient from the wave field with weak wave signals. Figure 6 shows the relationship between wavenumber deviations and $\widetilde{W}(\overrightarrow{\mathbf{x}})$, which represents the magnitude of wave signals for the analysis of the local phase gradient. The accuracy of wavenumber estimations is obviously poor, and the value of $\widetilde{W}(\overrightarrow{\mathbf{x}})$ is too low. In general, the higher the value of $\widetilde{W}(\overrightarrow{\mathbf{x}})$ is, the more reliable the results of wavenumber estimations.

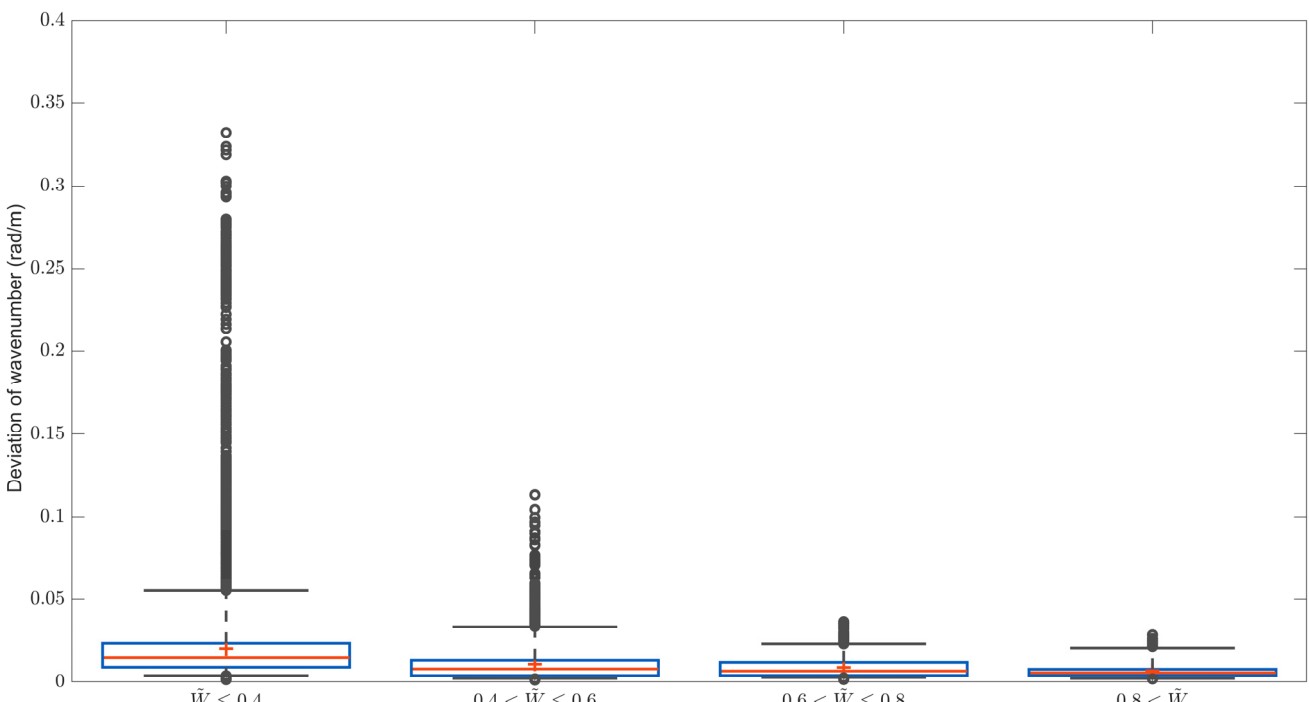

**Figure 6.** Relationship between the deviations in the wavenumber and $\widetilde{W}(\vec{\mathbf{x}})$. The definitions of different markers within the boxplot are the same as those in Figure 5.

After determining the best weighted least-squares fits of the dispersion relation curve from the estimated wavenumbers of different spatial locations $\vec{\mathbf{x}}$, we obtain the bathymetry result, which is shown in Figure 7a. The resolution of the estimated depth is set to 0.01 m. The relationship between the estimated wavenumbers and the fitted dispersion curve is also key to quantifying the accuracy of the depth determination. For each of the fitting results from different locations, we can estimate the coefficient of determination $R^2$ from the estimated wavenumbers and modeled wavenumbers of the fitted dispersion relation curve:

$$R^2 = 1 - \frac{\sum (k - k_m)^2}{\sum (k - \bar{k})^2}, \tag{16}$$

where $k_m$ is the modeled wavenumber of the fitted dispersion relation curve and $\bar{k}$ is the average value of $k$.

In general, $R^2$ is defined as the square of the correlation between the modeled wavenumber and the estimated wavenumbers $k$. However, $R^2$ can produce a negative value when the model poorly fits $k$ [24]. Figure 7b shows the results of $R^2$ in the spatial domain. Compared to the bathymetry map in Figure 7a, we observe that poor results of $R^2$ occur mostly in shallow-water locations. The defect of the linear dispersion relation in the determination of shallow water depth is obvious. Because our depth estimations are based on the fits of the dispersion relation curve using estimated wavenumbers, the accuracy of depth estimation should be related to $R^2$.

As shown in Figure 8, all the cases with negative values of the coefficient of determination are from locations where the in situ depths are less than 2 m. Wave nonlinearity, which is unavoidable in shallow-water areas, makes the wavenumber–frequency pairs deviate clearly from the linear wave dispersion relation curve and makes the depth estimation unreliable. Compared to the errors of depth estimation from shallow-water areas, the relationships between $R^2$ and depth errors from deeper-water areas show clearer trends. To determine these trends, the linear regression lines of different categories are also presented in Figure 8. The relations between $R^2$ and the errors of depth estimations from deeper-

water areas of radar images are obvious. The lower the values of $R^2$ are, the higher the errors of depth estimation from deeper-water areas. We can confirm the quality of depth estimations according to the value of $R^2$. Figure 9 shows the comparisons of the depth estimations between the radar-estimated depth and survey depth. For some safety-related applications, e.g., ship navigation, depth information with poor accuracy can be worse than insufficient depth information. In consideration of the quality of depth estimations, we remove the estimated depth results whose $R^2$ values are less than 0.6.

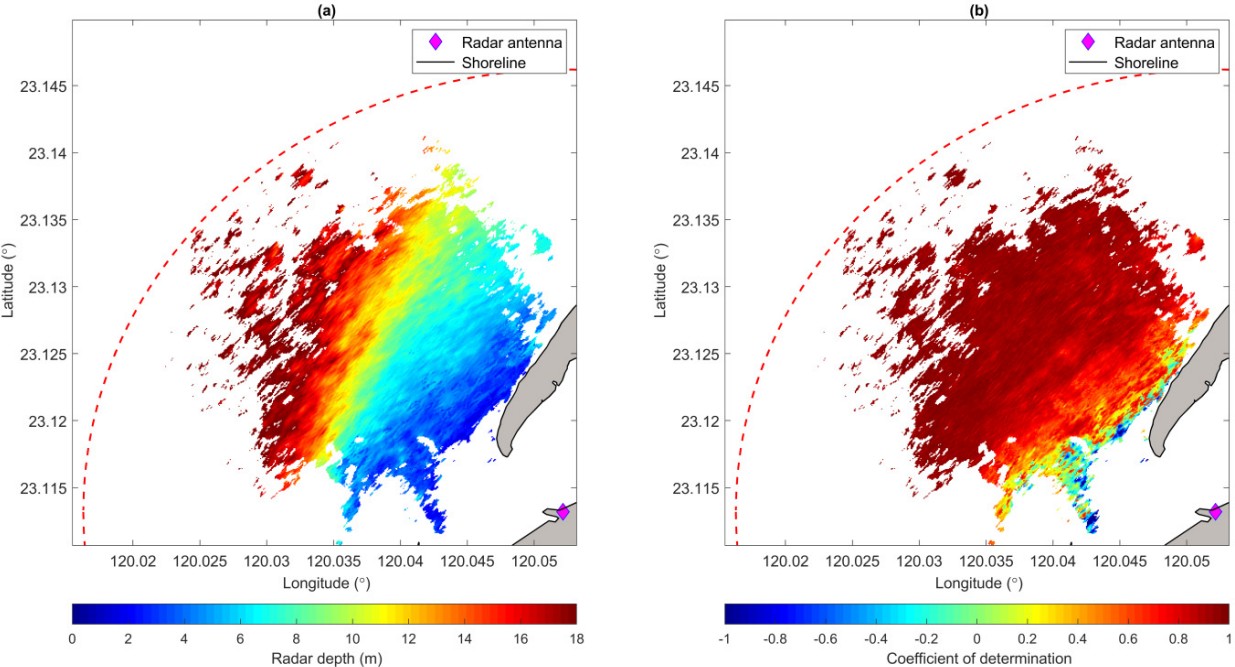

**Figure 7.** (**a**). Bathymetry results are estimated from radar images. (**b**). Coefficient of the determination results between the estimated wavenumbers and the fitted dispersion relation curve.

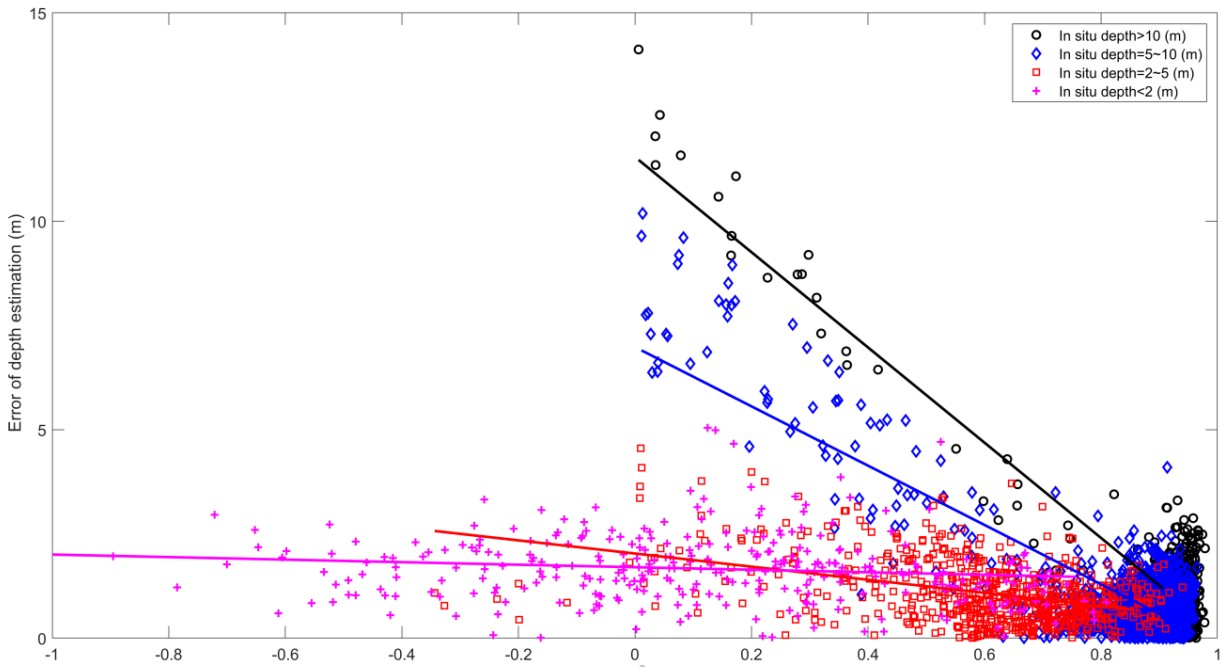

**Figure 8.** Relationship between the coefficient of determination and the error of depth estimation. The linear regression lines for each point category are also shown in this figure.

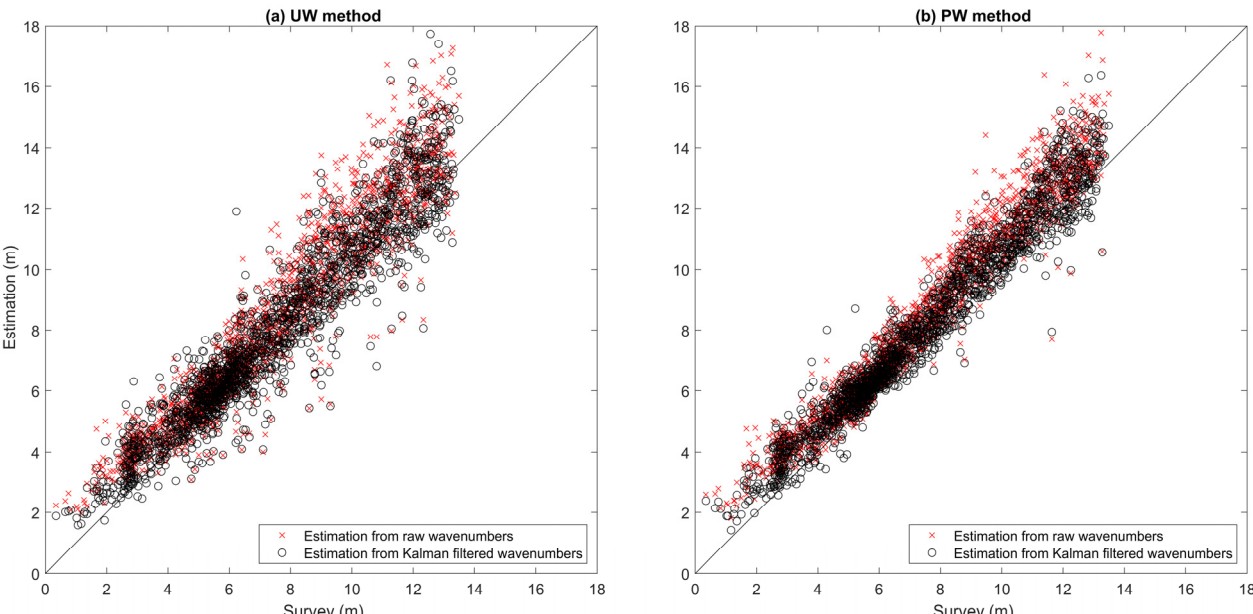

**Figure 9.** Scatter plot of the water depth determined from the radar images and the survey data. (**a**). UW method: A single wavenumber from each frequency bin is applied to fit the depth. (**b**). PW method: Plural wavenumbers from each frequency bin are applied to fit the depth.

Although the depth results from Figure 9a,b are all obtained by the least-squares fit, the depth estimations are obtained using different ways of estimating the wavenumbers. For the UW method in Figure 9a, there is only one wavenumber result from each frequency bin. The wavenumber is estimated from the nearly monocomponent wave field, which implements only directional filtering with $\theta = \theta_p \pm (\Delta\theta/2)$ from Equation (3). However, the PW method in Figure 9b shows the results from abundant wavenumbers that are implemented using different $\theta_j$ from Equation (3). For each of the methods, we also discuss the depth estimations with and without the Kalman filter. Both Figure 9a,b show that the Kalman-filtered wavenumbers are helpful for obtaining more reliable depth results. Compared to the results of Figure 9a, the depth estimations using the least-squares fit of the plural wavenumber–frequency pairs can be improved. To evaluate the accuracy of depth estimation using estimated wavenumbers after Kalman filtering, we estimate different statistical parameters from the estimations and ground truths shown in Table 1. Although the correlation coefficients of different methods are all over 0.95, the depth estimations using abundant wavenumbers still show a higher correlation with the survey depth. In addition, the root-mean-square deviation of depth estimation can be improved when the Kalman-filtered wavenumbers are applied to estimate the depth.

**Table 1.** Statistical results of depth estimations.

|  | UW Method | | PW Method | |
|---|---|---|---|---|
|  | Raw Wavenumbers | Kalman-Filtered Wavenumbers | Raw Wavenumbers | Kalman-Filtered Wavenumbers |
| Correlation coefficient | 0.95 | 0.95 | 0.97 | 0.97 |
| Root-mean-square deviation | 1.43 | 1.17 | 1.26 | 0.98 |
| Slope of the regression lines | 1.00 | 0.97 | 1.03 | 0.99 |
| Bias | 0.91 | 0.81 | 0.72 | 0.73 |

## 4. Limitations and Uncertainties

An accurate wavenumber is undoubtedly necessary to obtain the reliable depth result using wave dispersion relation. Figure 5 shows that most of the inaccurate wavenumber estimations are derived from higher-frequency bins. According to Equation (1), the influence of depth on the wavenumber–frequency pairs is obvious at lower frequencies. In contrast, the wave–current interaction is obvious at higher frequencies. The coastal current can be highly nonstationary and inhomogeneous due to the influences of sea states and coastal bathymetry. In our study, the wave–current interaction is assumed to be neglected when using the dispersion relationship. However, the influences of sea surface currents induce wavenumbers in high-frequency bins to deviate from the dispersion relation curve without currents. Another reason for the inaccurate wavenumber is the aliasing effect. In the physical sciences, the wavenumber is the spatial frequency of a wave. The complex spectral densities at higher frequencies are easily influenced by aliasing [29]. Due to the overlap of complex spectral densities at spatial frequencies near the Nyquist spatial frequency, the influence of phase estimation from complex spectral densities is unavoidable.

The scatter plot in Figure 9 shows that the Kalman-filtered wavenumbers are helpful for obtaining more reliable depth results. However, the accuracy of radar estimations seems to depend on the survey depth. Figure 10 presents the differences between radar depths and survey depths, which are classified into different conditions of survey depths. We observed that most of the radar-estimated depths are overestimated from the shallower area. As we pointed out in the data analysis section, the wavenumber–frequency pairs at higher-frequency bins, whose normalized magnitude is $\widetilde{W}(\vec{\mathbf{x}})$, are mostly weak and are excluded from the fitting process. The nonlinear effects are often unavoidable for the wavenumber–frequency pairs at lower-frequency bins, even though we removed the estimations whose $R^2$ values are too low. Large amplitude waves in shallow water are known to travel slightly faster than predicted by linear wave theory, which causes a slight overestimation of the radar depth [30]. This finding should explain the positive bias of radar-estimated depth estimation. The cases of shallow-water areas are often overestimated, regardless of whether Kalman filtering is applied.

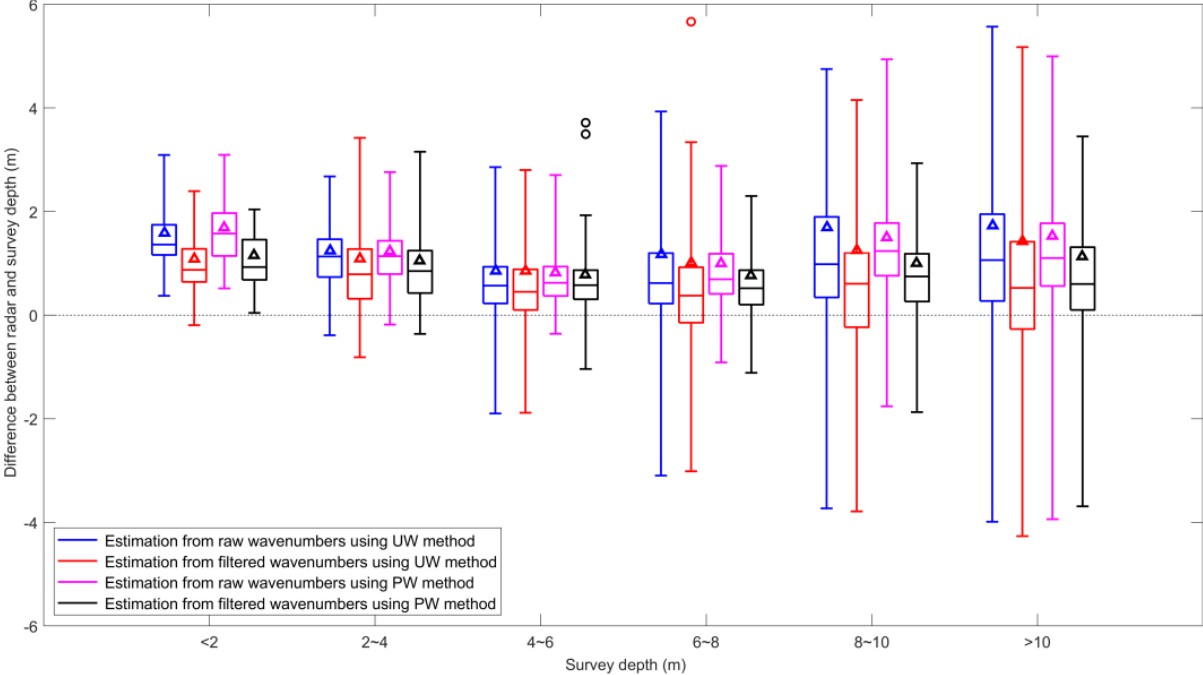

**Figure 10.** Box plot of the water depth determined from the radar images and survey data. The definitions of box, whisker, and outlier are the same as those in Figure 5. The triangle markers are defined as the root-mean-square deviation of each category.

The overestimation of radar-estimated depths is not obvious in deeper water. However, we observe more inaccurate radar-estimated depths from the sea area where the survey depths exceed 10 m. The uncertain radar estimations in the deep-water case are related to the unclear wave patterns on the radar image. In our study site, the deep-water areas are located farther from the radar antenna. Because the radar echo intensities decay with distance, the unclear wave pattern on the radar image is unavoidable. A sufficiently clear wave pattern is important for estimating the wavenumbers.

## 5. Conclusions

The phase gradient (PG) has been demonstrated to be a powerful tool for extracting local wavenumbers from spatial images. Since only one local PG can be found along the horizontal or vertical direction, only one wavenumber modulus can be obtained at any given spatial location of a remotely sensed image. Although it is still possible to use a single wavenumber to estimate depth using the theory of the wave dispersion relation, the wavenumber obtained from the wave field of a remotely sensed image is sensitive to noise. The depth result can be uncertain if a single and unreliable wavenumber is used. Some studies obtain the representative wavenumber, which can be more reliable than the wavenumber from the PG method, i.e., from the 2D image spectrum. However, the resolution between the spatiotemporal domain and the wavenumber–frequency domain is the opposite of that based on the Heisenberg uncertainty principle. To obtain reliable and accurate wavenumber and frequency results from the spectrum, degeneration of the spatiotemporal resolution is unavoidable.

The depth determination in our study is based on the fit of the dispersion relationship curve to abundant frequency-dependent wavenumbers. Some previous studies extracted different wavenumber–frequency pairs from different frequency bins of the image spectrum and then best fitted all frequencies to determine the local depth. Under the same idea, we extract the features of the plural wavenumbers of each frequency bin from fixed locations within the wave field image. To extract more wavenumbers of each frequency bin from the same location, we applied directional filtering technology. Different nearly monocomponent wave images can be produced from remotely sensed images using different directional pass filters. As a result, we can estimate more wavenumbers from nearly monocomponent wave images using the PG method.

The remotely sensed images we analyzed in this study are X-band radar images. Within the area of radar monitoring, we have the in situ depth data that were measured 8 days after the radar observation. Due to the influences of ocean currents and image noise, the raw wavenumber results extracted from the nearly monocomponent wave images are highly uncertain in higher-frequency bins. The Kalman filter is confirmed to improve highly scattered wavenumbers. For the cases with lower coefficients of determination, the errors of depth estimation from deeper-water areas are higher. As a result, the coefficient of determination can be used as an index to obtain reliable depth estimations. After removing the depth estimations with low coefficients of determination, we discuss the performance of radar depths using the survey depth data. In contrast to the depth estimations from a single wavenumber of each frequency bin, the depth estimations from more wavenumbers of each frequency bin show results that are more reliable. In addition, Kalman filtering is helpful in reducing the root-mean-square deviation of depth estimations.

As stated before, the spatial resolution of bathymetry determination using the PG method can be higher than that using the spectral method. In previous studies, the application of spatial or temporal filters was common in reducing the error of bathymetry maps. In our study, there is no low-pass filter for the spatial domain in the process of depth determination from our radar images, and the spatial resolution of the bathymetry maps is the same as that of radar images. Long-term radar image sequences are also unnecessary for our image process. After checking the quality of the depth results using the coefficient of determination, 128 continuous radar images that take approximately three minutes were found to be sufficient to obtain a reliable bathymetry map. We also confirm

that directional filtering can produce different nearly monocomponent wave images and then obtain plural local wavenumbers at the same location. Plural wavenumber fitting is workable in improving the reliability of depth estimation.

**Author Contributions:** Writing—original draft, L.-C.W.; Writing—review and editing, L.Z.-H.C., L.-C.W., Y.-D.S. and J.-W.L.; Conceptualization, L.Z.-H.C., L.-C.W. and J.-W.L.; Data curation, L.Z.-H.C.; Formal analysis, L.-C.W.; Funding acquisition, L.Z.-H.C.; Investigation, L.Z.-H.C. and Y.-D.S.; resources, Y.-D.S.; Methodology, L.-C.W. and L.Z.-H.C.; Supervision, L.Z.-H.C.; Visualization, L.Z.-H.C. and L.-C.W. All authors have read and agreed to the published version of the manuscript.

**Funding:** This research was funded by the Ministry of Science and Technology, Taiwan under grants MOST 109-2623-E-006-002–D and MOST 109-2221-E-006-101.

**Institutional Review Board Statement:** Not applicable.

**Informed Consent Statement:** Not applicable.

**Data Availability Statement:** The data presented in this study are available upon request from the corresponding author. The data are not publicly available due to security issues.

**Acknowledgments:** The survey bathymetry data are provided by The 6th River Management Office of Water Resources Agency, which is a governmental agency in Taiwan.

**Conflicts of Interest:** The authors declare no conflict of interest.

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
