# Peer review of "Bathymetry Determination Based on Abundant Wavenumbers Estimated from the Local Phase Gradient of X-Band Radar Images"

_remotesensing, doi:10.3390/rs13214240_

Round 1

Reviewer 1 Report

Your paper deals with a topic of strong inteset to the community, especially in the filed of temporal seabed changes (local area). The paper is well written and pleasant to read.

The main worry that I have at this level is that at this level concerns the lack of information concerning the reference survey (single beam). As a matter of fact, you should provide more details concerning the vertical reference (and its accuracy) of each of the comparing data sources. By simply looking at Fig 9, it appears that they are a biais (positive towards the estimations compared to the survey - point cloud appears above the y=x relation).

Reviewer 2 Report

The article represents a bathymetry determination based on a phase gradient methodology. The method was adequately described but since the authors also collected the acoustic datasets, I strongly recommend improving the paper by evaluating and comparing radar bathymetry with results from singlebeam echosounder survey. How does the accuracy of one data compare to the other?

Some specific comments:

line 42: references needed. Consider e.g.:

  • Guinan, J.; McKeon, C.; O'Keeffe, E.; Monteys, X.; Sacchetti, F.; Coughlan, M.; Nic Aonghusa, C. INFOMAR data supports offshore energy development and marine spatial planning in the Irish offshore via the EMODnet Geology portal. Quarterly Journal of Engineering Geology and Hydrogeology 2021, 54, doi:10.1144/qjegh2020-033.
  • Snellen, M.; Gaida, T.C.; Koop, L.; Alevizos, E.; Simons, D.G. Performance of Multibeam Echosounder Backscatter-Based Classification for Monitoring Sediment Distributions Using Multitemporal Large-Scale Ocean Data Sets. IEEE Journal of Oceanic Engineering 2019, 44, 142-155, doi:10.1109/joe.2018.2791878.
  • Janowski, L.; Wroblewski, R.; Dworniczak, J.; Kolakowski, M.; Rogowska, K.; Wojcik, M.; Gajewski, J. Offshore benthic habitat mapping based on object-based image analysis and geomorphometric approach. A case study from the Slupsk Bank, Southern Baltic Sea. Science of The Total Environment 2021, 801, 149712, doi:10.1016/j.scitotenv.2021.149712.

line 50: What about the resolution and accuracy of radar images in coastal areas compared to underwater acoustics?

line 223: provide the manufacturer and specific device of singlebeam echosounder. If possible, provide also resultant bathymetry profiles from this device. 

line 226: clarify the source of the survey bathymetry

Reviewer 3 Report

Dear Authors,

The research results are interesting and have great practical potential. I have some recommendations for improving the content of the manuscript.

  • In the Keywords, there is no need to use the same words and phrases as in the title of the manuscript, as this reduces their effectiveness.
  • The structure of the manuscript needs to be changed. I propose the following structure:
  1. Introduction
  2. Data and Methods
    • Study area
    • Theoretical Preliminaries
    • Data and Their Sources
    • Methods
  3. Results and Discussion
  4. Limitations and Uncertainties
  5. Conclusions
  • Figure 8. It is desirable to plot the most optimal trend lines for each point category.
  • The section "Conclusions" should contain only the most essential information (findings). The rest needs to be moved to the new section "Limitations and Uncertainties".

Round 2

Reviewer 2 Report

My comments were addressed properly. I would like to see this paper published.

Reviewer 3 Report

Dear Authors,
Thanks for the edits made in the manuscript. I have one small recommendation left.
Figure 8. Earlier I asked you to draw trend lines in this figure. You did it. Would you please show the coefficient of determination (R2) for each line?
  Thanks.